# Hierarchical Bayesian Modeling for Clustering Sparse Sequences in the Context of Group Profiling

## Abstract

This paper proposes a sparse sequence clustering algorithm based on hierarchical Bayesian Modeling. This algorithm can cluster very sparse non-negative integer sequences of unequal length which do not necessarily belong to well-defined classes. Such Sequences are generated from the vast majority of normal human actions, for example, user behavior data for Wikipedia contributors expressed as a count of updates per day. Thus this algorithm and modeling technique is very useful for modeling non-negative integer sequences generated by real-life human actions, such as user-visits to websites or shops. This data model is a mixture model where every sequence is generated by a mixture of distributions associated to several clusters but does not need the data to be represented by a Gaussian mixture, which is the most commonly used representation of sequences and that gives significant modeling freedom. This algorithm also generates a very interpretable profile for the discovered Clusters,i.e, the latent groups of users who visited a food joint over a two year period. The Cluster profile, in this case, would contain the representative visit-counts of the cluster or group. The data that was used for the work has been contributed by a digital marketing company. The data is a collection of sparse sequences of a number of user visits to a food joint, where each entry of a user's sequence is the number of visits of that user per week to a given restaurant.

## 1 Introduction

Most of the existing sequence classification or clustering techniques make some assumptions which are not applicable for sequences that are generated from real-world human actions and that can be represented as a chain of events. Some examples of such sequences are following: the sequence of counts of weekly user-visits to a food-joint collected for a several weeks period, click-stream data of user navigation, the order in which the items are added to a shopping cart, the sequence of weekly updates of the Wikipedia users. All these sequences show a common property; they are a sequence of non-negative integers. However, most of the existing sequence modeling techniques assume that the entries of the sequences are real numbers. For example, in the worlds largest collection of time series data-sets, the UCR classification archive, all forty-five time-series data-sets are only real number sequences.

Other than this real number sequence assumption,most of the existing sequence classification techniques make some more assumptions implicitly or explicitly as described in Bing Hu's paper (**?**)"To summarize, much of the progress in time series classification from streams in the last decade is almost certainly optimistic, given that most of the literature implicitly or explicitly assumes one or more of the following: 1. Copious amounts of perfectly aligned atomic patterns can be obtained [11][35][37]. 2. The patterns are all of equal length [11][13][16][23][31]. 3. Every item that we attempt to classify belongs to exactly one of our well-defined classes [10][13][23][30]." (et.al., 2009) (E.Keogh) (Koch, 2010) (Prkk, 2006) (Ratanamahatana & Keogh, 2004) (Reiss & Stricker, 2011) (D.GafurovandE.Snekkenes, 2008) These three assumptions are very common in existing works about sequence modeling. However, in this paper, we consciously avoid making these assumptions, since these assumptions are not applicable in real world count data. For example, in this work, the data that has been used are sequences of counts of weekly user-visits to a food-joint

collected for a two year period. But the beginning of the sequence of each user is not at the user's beginning of her lifetime in the business and the end of the sequence is not the end of the user's lifetime. Each user's sequence starts and ends at different phases of their lifetime in the business. Thus the sequences that have been used in this work are not aligned. Also, the sequences are not of equal length in this study, but their maximum length is 104 (weeks). There is no way to classify these users into well-defined classes other than hand annotating them, which is an enormous task and impractical to do in a business environment. Moreover, there are no well-accepted criteria to define the classes of the users which can be taken to be a gold standard for evaluation.

One more feature of the real world sequences that are generated from human actions and more specifically, user behavior, is sparseness. For example in this weekly user-visits data that has been used for the work, most of the weekly visit entries of most of the users are zero, because most of the users of this business are infrequent visitors. This is another challenge that has been addressed in this work. Since the data is so sparse, or in other words, the sequences mostly contain zeros and it is difficult to be certain about a structure in the data. Thus we did not model the sequence as a time series model like Markov chain where each entry is dependent on its previous entry. We modeled the sequence as a bag of words model being generated from a mixture of distribution because it can provide us with much more flexibility of modeling.

Daily human activities are determined by real-life motivators such as schedules, locations, tastes, and expectations. The similarity in the motivators can give rise to the similarity in behavior. Thus, the similarity in motivators can create latent groups among users who visit a shop or a restaurant or a website. In this work, an algorithm is proposed to identify such latent groups and to generate a group profile for the latent group. An individual user is perceived as partially affiliated to every latent group and from the data, the proportion of the user affiliation to every latent group is inferred. As we model the users to be soft assigned to every latent group with a proportion of affiliation to the groups, our mixture model can accommodate users that do not belong to a well-defined cluster. It is assumed that this mixing proportion or affiliation proportion is constant for the whole user sequence. User-group profiles are generated through clustering the user behavior sequence data using Hierarchical Bayesian Modeling and then by inference of the model parameters. A function of each of the cluster-specific parameters in the model constitutes each entry of each latent group profile.

In this work, we assume a generative model where each individual sequence is generated by a Poisson distribution whose parameter is a convolution of K parameters drawn from K distributions associated with each of the K clusters. The entries of the sequence are assumed to be independent of the entries before or after it given the parameters of the generating distributions. Thus, even if the data is sequential in time, it has not been modeled as a time series sequence, which could be done using something like HMM. The reason for doing this is following. Primarily, each user sequence does not span the whole lifetime of that user in the loyalty system but is arbitrarily selected two-year span in the users lifetime. So unlike RNA sequencing problems, we do not know the whole sequence and so we do not have enough data for assuming a structure in the part of the sequence that we have. Secondly, the data that has been used for this work has been aggregated into weeks from the daily user visit data in a restaurant for two years. If every daily entry is treated as a feature of the sequence, the number of the feature of each sequence is 730. Moreover, the data is very sparse. So we have a large number of features whose values are mostly zero. Thus the data do not provide enough support for making any assumptions about the dependence structure among the features. The assumption of independence is often made for high-dimensional continuous data (T Speed, 2001), (Tibshirani, 2003), (Witten, 2011) since for high dimensional data there are too few observations available to be able to effectively estimate the dependence structure among the features. Thus we assume that the sequence of each user is generated by a mixture of distribution, but not assuming the direct dependence of a later entry in a sequence on a former entry in the sequence.

In this work, we use a Bayesian Finite mixture model where the number of mixture components or the number of clusters we provide beforehand, the number of clusters being determined empirically. Bayesian finite mixture models require the specification of the number of mixture components, or groups, a priori. Existing examples assume the number of groups to be unknown and therefore most commonly infer this prior specification based on marginal likelihoods or Bayes factors (J Lau). For example, (Fahey) used the Bayesian Information Criterion to determine the optimal number of groups when analyzing dietary patterns based on the UK National Diet and Nutrition Survey. Again,

this approach often leads to complex group narratives, for example, (Fahey) identified six groups, with narratives differing intricately in their consumption of multiple different food types.

Conversely, Bayesian infinite mixture models assume an infinite number of mixture components, with the total number of occupied groups inferred from the data during model fitting (S Kim & Vannucci). This approach often leads to a large number of groups containing a small percentage of individuals, complicating group narratives further. For example, using this approach, (A Crpet, 2011)) identified seventeen groups within the Individual and National Study on Food Consumption, differing in dietary consumption behaviour, with only three of the groups accounting for 98 percent of the surveyed individuals. Similarly, (Swartz) employed this approach to group students based on surveyed course satisfaction, identifying 10 groups within just 75 surveyed students, with many groups differing only slightly in their satisfaction response profiles.

The existing Sequence Clustering Techniques follow variations of three basic ideas. In several bio-informatics papers on gene-sequence clustering is based on non sequence attributes, such as the length of the sequence and similarity of the entries are considered to be the main features based on which the clustering is done. This technique is not a model based technique. Also as variable length sequences are not feature vectors, the 'distance' between sequences is an ambiguous term, which should be defined clearly in context.Based on the idea of distance the 'similarity' between sequences is measured, thus this measurement needs to be defined too. The second technique is based on modeling sequences as Markov chains where the order of the chain determines number of past states affecting current state.As the order increases the cost of modeling also scales up in this model and also this model does not support the idea of an unobserved variable which might control the generation of the sequence, which is often the case in practical scenario. The third technique is based on hidden Markov model where the observed outcome is a resultant of hidden latent state.This papers presents a hierarchical Bayesian model for soft clustering of the sequences. The proposed technique is ideologically closest to an HMM, since HMM can be considered a generalization of a mixture model where the hidden variables (or latent states), which control the mixture component to be selected for each observation, are related through a Markov process. This model however is more interpretable in terms of its parameters associated to each sequence and the clusters/groups of sequences.Thus the parameters and some quantities derived from the parameters can be mapped to real physical quantities and can be used to create group profiles.

## 1.1 BUSINESS SCENARIO

The data used in this paper is real industrial data from a loyalty program where by using an enrolled service people can earn points or rewards.In a loyalty program users enroll, use the service through the loyalty program for a period and then at some point become disinterested in using the program and thus churn out of the pool of the users of the loyalty program.It is important to study user behavior to predict his next visit, predict probability of churning out and study his buying behavior. Not only in case of individual users, but identifying the latent groups and studying user-group behavior for next-visit, churn-probability and buying behavior is also important for user segment targeting in a campaign. In this work, we have modelled user visit behavior. The user visit-behavior is recorded as a sequence of number of visits to a restaurant every week. Hence all the users are represented by variable length sequences where every entry of a sequence is the number of visits of the corresponding user in a particular week.

## 2 LITERATURE REVIEW

In this paper we describe sequence modeling, latent group identification from user-visit sequence and clustering as a result of sequence modeling. Hence the Literature review section has been divided into three sub parts: Discovering groups among users, Clustering techniques in general and Sequence clustering.

## 2.1 DISCOVERING GROUPS AMONG USERS

The existing literature related to user group profiling does not necessarily look for latent groups that naturally existed among the users, but somehow manages to put some users together as a group for eliciting a preference of the group of users for certain product or item given the data about individual

user's preference for that product or item (Senot, 2010). For example in a rating based system, the objective of the existing literature is to generate a group rating for an item from individual ratings for that item. In the existing literature, the problem has been approached in two different ways. One way is aggregating the ratings of the users' group and generating a single rating for the group. (Masthoff, 2002) Another way is to create a representative user for the user group and predicting the rating by that representative user for the given item. (McCarthy, 2006) (OConnor, 2001)

The group of users thus selected by the existing literature lack certain characteristics. First, these people may not really be connected in any intrinsic manner, they are apparently just random people thrown together. For example, MUSICFX (McCarthy, 1998) selects music channels for the music to be played in a fitness center. Based on the preferences that have been previously specified by the members who are currently working out, the system chooses one of 91 possible music channels, including some randomness in the process.

INTRIGUE (Ardissono, 2003) recommends tourist attractions for heterogeneous groups of tourists that include relatively homogeneous subgroups (e.g. children).

However, the people working out on a particular day and time has no connection between their tastes in music! Similarly, random tourists making a tour plan have no inherent connection. Hence, they share no commonality that a group should share. In this paper, a group is a latent group, sharing similar behavior pattern. Latent groups are discovered by hierarchical Bayesian modeling of the user-visit data. Hence, they are more connected as a group and thus a more fit candidate for group profiling.

In case of a social network based user base, the existence of social groups is investigated (Richter Y, 2010) for estimating the probability of churning out of closely connected users from a loyalty program. The group among the users has been determined based on their social connected-ness. This process does use an intrinsic connection to define a group, but it is doubtful whether the churn behavior is at all impacted by social connection. In this paper,based on user behavior data we aim to find latent groups and to create a group user profile, which is a novel idea compared to these papers.

## 2.2 CLUSTERING TECHNIQUES

Clustering techniques can be divided into two groups based on their style of associating an item to a cluster: hard assignment and soft assignment.The algorithm proposed in this paper uses soft assignment technique. On the other hand K-means makes a hard assignment of the data points to the cluster depending on the euclidean distance. EM Clustering assumes that the data points are generated by a mixture of Gaussians and determines the probability of a data point being generated by a particular Gaussian (assigned to a cluster). Application of EM clustering is not possible if the data is not generated by a mixture of the Gaussian distributions. So, EM clustering cannot be applied to a data-set in which every data point is a sequence of zeros and positive numbers and cannot be represented as Gaussian. Our algorithm is conceptually similar to EM in the sense that effectively it creates a mixture model.But it gives the freedom that the distributions need not be Gaussian. Hierarchical Bayesian Clustering (Heller & Ghahramani, 2005), a probabilistic clustering technique, where we do not have to define the number of clusters. This technique probabilistically allocates a data point to a cluster and merges two clusters when they are very close. But as our data is very skewed, all the cluster centers it has are close, so merging becomes very arbitrary. Another good approach is the information-theoretic approach to Clustering (Slonim N, 2005).

## 2.3 SEQUENCE CLUSTERING

Sequence is an ordered list of things or events. In particular to this work we are concerned about sequence of events which are ordered by time,i.e., time series sequences. Even if we are discussing time series sequences, for reasons discussed in the introduction, the entries of the sequence are assumed to be independent of each other. So the LDA algorithm is ideologically close to this algorithm. In LDA the documents are treated as a sequence and the topics are treated as the clusters. LDA associates each word of a document to one topic and models each document as a mixture of topics. In this work, not only a sequence of user is a mixture, but each entry of the sequence is generated by a mixture of distributions. Since unlike a sequence of words of a document, the user-visit sequence is very sparse, i.e., mostly the value of each entry is zero, modeling each entry as generated

by a single cluster is not as effective as it is in case of a document. Also in case of sparse sequence, Poisson distribution is a much better choice than a multinomial distribution which is used in LDA.

In general sequence classification or clustering techniques can be divided into two categories.The first technique is where the sequence itself is treated as a vector or its features are extracted and those features constitute a vector. Then some similarity measure is used to cluster those vectors. Sequence Clustering by similarity measures has been mostly applied to RNA or DNA sequences, the similarity measure being edit distance, hamming distance,longest common sub-sequence.Another way of representing and clustering sequences is done by suffix tree. (Xing, 2010)(Agarwal, 2014). A variant of this feature extraction technique is modeling the sequence as a collection of independent entries, generated by a Poisson distribution or multinomial distribution and then computing a dissimilarity matrix. A RNA sequence clustering model was proposed using Poisson model [DM Witten]. [Berninger et al. (2008)] propose a method for computing a dissimilarity matrix using sequencing data. They assume that each observation is drawn from a multinomial distribution, and they test whether or not the multinomial parameters for each pair of observations are equal. [Anders and Huber (2010)] propose a variance-stabilizing transformation based on the negative binomial model, and suggest performing standard clustering procedures on the transformed datafor instance, one could perform hierarchical clustering after computing the squared Euclidean distances between the transformed observations.

This above technique of treating sequences as vectors is not applicable to our work. Firstly, the length of the sequences are unequal. Secondly, the entries does not represent any feature as the entries in a vector does.

The second technique of sequence modeling is using the structural information of a sequence and model it such that the generation of a later entry is dependent on former entries. These are probabilistic models, most common among them is built using HMM, of which one of the best papers is (Smyth, 1997).Microsoft paper uses Markov chains to model and cluster click user navigation pattern (Cadez, 2000). The following link discusses Microsoft's model for sequence clustering. (Microsoft, 2017)

There are many variations of sequence modeling, but the basic theme centres on the discussed techniques.

## 3 HIERARCHICAL BAYESIAN CLUSTERING MODEL

### 3.1 DESCRIPTION

Consider a data set $D$ consisting of $N$ sequences, $D = \{Y_1, Y_2, \ldots, Y_N\}$ and $Y_i = \{y_{i1}, y_{i2}, \ldots, y_{iL}\}$ is a sparse sequence of length $L_i$ of small positive numbers.The problem addressed here is to find $K$ latent groups in the data. The data used is described in detail in the business scenario section.$N$ sequences are sequences of restaurant visits of $N$ users. Thus $y_{iw}$ is the number of times the $i^{th}$ user visited the restaurant in the $w^{th}$ week. Since the number of user-visits every week is either zero or a small positive number,Zero being the most common entry,we assume that every user-visit sequence $i$ is generated from a Poisson distribution with mean-visits $\lambda_i$. We further assume that the user-visit sequences are generated from the mixture of $K$ distributions associated to $K$ clusters with distribution means $\eta_k$ s. $\eta_k$ is drawn from a gamma distribution with parameters $shape_k$ and $rate_k$. Every user is affiliated to every cluster and the affiliation proportion of user $i$ to cluster $k$ is $\alpha_{ik}$. $\alpha_{ik}$ is drawn from a Beta distribution and normalized, so that the sum of $\alpha_{ik}$s over $K$ clusters is 1. Every cluster has a contribution to the mean-visit $\lambda_i$ of user $i$. The contribution of the cluster $k$ to $\lambda_i$ is $\lambda_{ik}$.$\lambda_i$ is a convolution of all the $\lambda_{ik}$. $\lambda_{ik}$ is generated from a gamma distribution with distribution mean $\eta_k * \alpha_{ik}$ . The reason of this design choice is that we believe that the mean visit of cluster $K$ is going to impact the mean-visit of user $i$ proportionally to user $i$'s affiliation, $\alpha_{ik}$, in the kth cluster. For example, if for a user sequence $\alpha_{ik}$ is 1 for cluster $k$, then the mean of the distribution generating $\lambda_{ik}$ is $\eta_k$ and all other distributions have mean zero. That essentially means $\eta_k$ is very close to $\lambda_{ik}$, which is effectively same to $\lambda_i$ and hence the user completely belongs to the cluster $k$. However, for some user who is not strongly affiliated to any cluster, $\eta_k$s are sort of mixed in proportion $\alpha_{ik}$ to make $\lambda_i$. Thus mixture proportion $\alpha_{ik}$ is a measure of proportional impact of cluster mean $\eta_k$ on mean user-visits $\lambda_i$ of every user-sequence $i$.

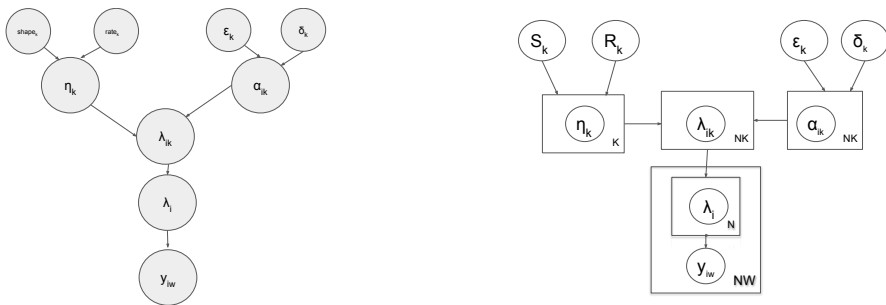

Figure 1: Group Profile Creation Procedure

## 3.2 MODEL

$shape \sim$ uniform (2,5)
$rate \sim$ uniform (0.1,1.0)
$\epsilon_k \sim$ uniform (0.1,1.0)
$\delta_k \sim$ uniform (0.1,1-$\epsilon_k$)
$\eta_k \sim$ gamma($shape$ , $rate$)
$\alpha_{ik} \sim$ Beta($\epsilon_k$,$\delta_k$)
(Normalized over k)
$\lambda_{ik} \sim$ gamma($\eta_k$ , $\alpha_{ik}$)
$\lambda_i = \sum \lambda_{ik}$
$y_{iw} \sim$ Poisson($\lambda_i$)

As we see from the model $\lambda_{ik}$ is the contribution of the distribution of cluster K to the user-visit-mean $\lambda_i$.The mean of the distribution of $\lambda_{ik}$ is $\eta_k$*$\alpha_{ik}$ i.e., cluster-mean multiplied by mixture proportion.

## 3.3 ALGORITHM

This algorithm assumes that number of clusters is known.The number of clusters K can also be estimated by calculating log-likelihood of the model for different values of K and by maximizing log-likelihood we can find a most possible K.

For every cluster k, we choose hyper-parameters shape, rate, $\epsilon_k$ and $\delta_k$ from uniform distribution. For each cluster k parameter $\eta_k$ is the representative value for number of restaurant visits for cluster k. $\eta_k$ is drawn from a gamma distribution with hyper-parameters $shape$ and $rate$. So the mean of the distribution of $\eta_k$s is $shape/rate$ and variance is $shape/rate^2$. Every cluster k has a parameter $\alpha_{ik}$, the mixture proportion which has a beta distribution with hyper parameter $\epsilon_k$ and $\delta_k$. So, $\epsilon_k/(\epsilon_k + \delta_k)$ is the mixture proportion averaged over the users for cluster $k$.This is an estimate of mean proportional user affiliation to cluster $K$. The algorithm derives the user group profile as a two dimensional vector where the entries are $\eta_k$, visit frequency of the user group and $\epsilon_k/(\epsilon_k + \delta_k)$, mean mixture proportion (user affiliation) to group $K$.

After $\alpha_{ik}$ is drawn it is normalized over k values for each User i. $\lambda_{ik}$ is drawn from a gamma distribution with parameter $\eta_k$ and $\alpha_{ik}$. $\lambda_{ik}$ is the contribution of the cluster k to $\lambda_i$, the expected number of weekly visits of user i. The distribution of $\lambda_i$ is sum of distributions of $\lambda_{ik}$s. As $\lambda_{ik}$ is drawn from K gamma distributions , $\lambda_i$ will also have a gamma distribution with $\phi$ and $\theta$ where $\phi = \sum(\alpha_{ikinv} * \eta_k)^2 / \sum(\alpha_{ikinv}^2 * \eta_k)$
$and$
$\theta = \sum(\alpha_{ikinv} * \eta_k)/\phi$

by approximation proposed by WelchSatterthwaite equation (Satterthwaite, 1946). $\alpha_{ikinv}$ is $1/\alpha_{ik}$. So we draw $\lambda_i$ from the approximate distribution stated above and use it as the parameter of the Poisson distribution of user visit frequency per week.

Table 1: Group profile for real data

| cluster | $\eta$ | $\epsilon_k/(\epsilon_k + \delta_k)$ |
|---------|--------|--------------------------------------|
| 1 | 0.317743 | 0.470588 |
| 2 | 0.640771 | 0.895 |
| 3 | 0.739492 | 0.46980 |
| 4 | 0.853064 | 0.47456 |
| 5 | 1.07128 | 0.34046 |

From the posterior of this model we sample $\eta_k$, $\epsilon_k$ and $\delta_k$, as they are our parameters of interest.

## 4 EXPERIMENTS AND RESULT DISCUSSIONS

### 4.1 DATA DESCRIPTION

The data is collected from a marketing company which promotes loyalty programs for its client company. Each record contains user id, the total number of restaurant visits by the user, number of visits every week for 113 weeks and the interval between visits . In this paper, 'number of visits every week' is the user behavior that has been modeled. Almost 20000 in 25000 total number of users visited at most 7 times in 113 weeks. However,there are at least 100 users who visited almost every week.From the business perspective these 100 people are very important and cannot be treated as outliers. This skewness is the reason why a clustering technique which tries to get equal size clusters, such as K-means is ineffective because all the cluster centers merge to one. We have taken a less sparse subsection of the data for the experiment, but still our algorithm also is impacted by the skewness. Thus it is important to effectively initialize the algorithm and also it is important to prune it at certain step when all the cluster centers are not too close.

From the viewpoint of sequence modeling every User's data is a sequence made of zeros and positive numbers, each entry representing the number of user visits every week. The data is very sparse and skewed because very few users visited very frequently and most of the users visited below seven times in the whole observation period. There are huge number of zeroes in the user visit Sequences since there is no restaurant visits most of the weeks.For the experiment we have selected users who meet at least a visit threshold, empirically determined. Still for all users, zero is the value for maximum entries.

## 5 SYNTHETIC DATA FOR SANITY CHECK

Other than using the industrial data we used synthetic data for the experiment. Three sets of data generated from three distinct Poisson distributions with parameters 0.2, 0.7 and 1.5. The size of these three sets were 100(lambda=0.2), 70 (lambda = 0.7) and 30 (lambda = 1.5). The difference in size of data is maintained for imitating the skewed nature of the real data.The parameters of Poisson distributions were chosen to be small numbers so that the synthetic data would have a huge number of zeroes in it and thus replicate the sparse nature of the actual sequence data.

The number of clusters K is chosen to be three in the experiment to match the number of distributions used to generate the data. In this experimental setting, the goal was to see the performance of the algorithm in finding three distinct latent groups, when there actually existed three latent groups in the data.We can see that each of the group parameters $\eta_k$s have three distinctly separated values, which shows the model could clearly cluster the synthetic sequences generated from three different distributions into three well separated clusters.Thus the model could bring out the latent structure in the data.

Table 2: Group Profile for synthetic data

| cluster | $\eta$ | $\epsilon_k/(\epsilon_k + \delta_k)$ |
|---------|--------|------------------------------------|
| 1 | 0.326886 | 0.45398 |
| 2 | 0.420915 | 0.4483566 |
| 3 | 0.62439 | 0.51660901 |

Table 3: mixture proportion of example sequence

| cluster | $\eta$ | $\lambda_k$ |
|---------|--------|-------------|
| 1 | 0.326886 | 0.0206129 |
| 2 | 0.420915 | 0.000552983 |
| 3 | 0.62439 | 0.0208292 |

## 5.1 EXPERIMENTAL PROCEDURE

The model was implemented in Pystan. Pystan is a Bayesian inference platform where if a model is compiled and the data is input,the software generates posterior and samples it using No-U-Turn Sampler or NUTS, which sets adaptive path length in Hamiltonian Monte Carlo.

The objective of the experiment was to learn the parameters of the distribution of the clusters$\epsilon_k/(\epsilon_k + \delta_k)$ and $\epsilon_k/(\epsilon_k + \delta_k)$. We started using 1000 iterations, but the model was converging too quickly to form K very close clusters whose $\eta_k$ values are very close. This is because of the inherent structure of the data, where the latent clusters are widely varying in size.To handle this skewness we recorded the parameters of the model at 500 iterations and that is reported in the table 1.However, In case of the synthetic data we did not have to prune the process, since the data were not that highly skewed and also the data had a clear underlying group structure as the data were generated by three different distributions.

## 5.2 RESULTS

The values of the cluster level parameters are listed in Table 1 for industrial data and Table 2 for synthetic data. The clusters are distinguishable, but as the sequences are sparse, the effect of high number of visits in a few weeks does not change the expected number of visits in a sequence in a big way.Thus the mean visits of every sequence has a low value and thus all the cluster means has low values.

## 6 CONCLUSIONS

This work presents a hierarchical Bayesian model for clustering sparse non-negative sequences and it also generates profiles of the latent user groups. The function of the hyper-parameter $\epsilon_k$ and $\delta_k$ of Table2 represents mean user affiliation of Cluster $k$. $\eta_k$ represents the mean visit frequency of the Cluster $k$. The uncertainty in the parameter values can be estimated by the hyper-parameters.Synthetic data has been used to do a sanity check and the algorithm gives the expected outcome.

## 6.1 FUTURE WORK

For future work, this model is proposed to incorporate time-dependency in proportional affiliation for every user to every cluster. As we do not have sufficient information about the structural dependency between entries of the sequence, we do not model each entry as a function of another. Rather we assume that a later value of the mixing proportion might be correlated to its last value. To have a very flexible model of this correlation we design that a former value of the mixing proportion would the mean value of the distribution from which a later value of mixing proportion is drawn.

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
