# OpenReview forum: "Hierarchical Bayesian Modeling for Clustering Sparse Sequences in the Context of Group Profiling"
_ICLR.cc/2019/Conference_

### Official Review · AnonReviewer4 · 2018-10-30
**HIERARCHICAL BAYESIAN MODELING FOR CLUSTERING SPARSE SEQUENCES IN THE CONTEXT OF GROUP PROFILING**

**Rating:** 2
**Confidence:** 4

**Review:**

The paper discusses clustering sparse sequences using some mixture model. It discusses results about clustering data obtained from a restaurant loyalty program.

It is not clear to me what the research contribution of the paper is. What I see is that some known techniques were used to cluster the loyalty program data and some properties of the experiments conducted noted down. No comparisons are made. I am not sure what to evaluate in this paper.

---

### Official Review · AnonReviewer3 · 2018-10-30
**The authors discuss a hierarchical Bayesian framework for clustering sparse sequences. They use data from a restaurant loyalty program to identify users (rows) and weeks of visits (columns) under the assumption that user visits to a restaurant will be sparse across weeks.**

**Rating:** 1
**Confidence:** 5

**Review:**

The paper is very poorly written. It is hard to understand what the real contribution is in this paper.
The connection of the model with HMM is not clear. The literature review has to be rewritten.

To the reader, it sounds that the authors are confused with the fundamentals itself: mixture model, Bayesian models, inference.

> Mixture models can be based on any of the exponential family distributions - Gaussian just happens to be the most commonly used.
> Again if this is a Bayesian model, why are #clusters not inferred? The authors further mention that in their Pystan implementation K clusters were spun too quick. What was the K used here? Was it set to a very large value or just 3? Did the authors eventually use the truncated infinite mixture model in Pystan?
> The authors mention their model is conceptually similar to EM but then end up using NUTS.
> Why is a url given in Section 2.3 instead of being given in the references?
> Provide a plate model describing Section 3.2.

---

### Official Review · AnonReviewer5 · 2018-11-02
**Both the writing and experiments should be improved**

**Rating:** 3
**Confidence:** 4

**Review:**

This paper propose a hierarchical Bayesian model to cluster sparse sequences data. The observations are modeled as Poisson distributions, whose rate parameter \lambda_i is written as the summation of \lambda_{ik}, a Gamma distribution with rate equal to the mixture proportion \alpha_{ik}. The model is implemented in Pystan. Experimental results on a real-world user visit dataset were presented.

The format of this paper, including the listing in the introduction section, the long url in section 2.3, and the model specification in section 3.2, can be improved. In particular, the presentation of the model would be more clear if the graphical model can be specified.

The motivation of choosing the observation model and priors is not clear. In section 3, the author described the details of model specification without explaining why those design choices were appropriate for modeling sparse sequence data.

Experimental results on a real-world dataset is presented. However, to demonstrate how the model works, it would be best to add synthetic experiments as sanity check. Results using common baseline approaches should also be presented. The results should also be properly quantified in order to compare the relative advantage of different approaches.

---

### Official Review · AnonReviewer2 · 2018-11-03
**Zero novelty**

**Rating:** 2
**Confidence:** 5

**Review:**

The problem formulation at the bottom of page 3 correspond to what a bag of words preprocessing of a document would provide and in this the clustering would be a much simpler solution that just doing LDA.

The paper has zero interest.

---

### Official Review · AnonReviewer1 · 2018-11-05
**poorly written paper, not ready for publication**

**Rating:** 2
**Confidence:** 5

**Review:**

Pros:
-- Clustering sequence vectors is a practical and useful problem. Some of the business use-cases described in the paper are indeed useful and relevant for analytics in healthcare and retail.

Cons:
-- The paper is poorly written. There are numerous typos and grammatical errors throughout the paper.
-- The ideas are not presented coherently. The writing needs to improve quite a bit to get accepted at a conference like ICLR.
-- Description of related literature is done very poorly.
-- The generative model described clearly lacks justification. The model is not described concretely either. There is no clear description of the inference techniques used.
-- Empirical results are weak.

---

### Meta-Review · Area_Chair1 · 2018-12-14
**Meta-Review for Group Profiling paper**

**Confidence:** 5
**Recommendation:** Reject

**Metareview:**

All reviewers agree to reject. While there were many positive points to this work, reviewers believed that it was not yet ready for acceptance.